# Methods Used for the Compaction and Molding of Ceramic Matrix Composites Reinforced with Carbon Nanotubes

**Valerii P. Meshalkin and Alexey V. Belyakov \***

Mendeleev University of Chemical Technology of Russia (MUCTR), 9 Miusskaya Square,
125047 Moscow, Russia; vmeshalkin@muctr.ru
**\*** Correspondence: av_bel@bk.ru; Tel.: +7-495-4953866

**Abstract:** Ceramic matrix composites reinforced with carbon nanotubes are becoming increasingly popular in industry due to their astonishing mechanical properties and taking into account the fact that advanced production technologies make carbon nanotubes increasingly affordable. In the present paper, the most convenient contemporary methods used for the compaction of molding masses composed of either technical ceramics or ceramic matrix composites reinforced with carbon nanotubes are surveyed. This stage that precedes debinding and sintering plays the key role in getting pore-free equal-density ceramics at the scale of mass production. The methods include: compaction in sealed and collector molds, cold isostatic and quasi-isostatic compaction; dynamic compaction methods, such as magnetic pulse, vibration, and ultrasonic compaction; extrusion, stamping, and injection; casting from aqueous and non-aqueous slips; tape and gel casting. Capabilities of mold-free approaches to produce precisely shaped ceramic bodies are also critically analyzed, including green ceramic machining and additive manufacturing technologies.

**Keywords:** carbon nanotubes; ceramic matrix composites; compaction; molding; casting; powder mixtures; green bodies; plastic molding powders; slips; polymerizable monomers; solid freeform fabrication; machinery

## 1. Introduction

Compaction molding is an important technological stage in the mass production of technical ceramics and ceramic matrix composites (hereinafter, CMCs). This stage commonly starts with a mixture of powders or a slip (commonly referred to as a molding mass) and finishes up with a compacted green body, or green ceramic. It is followed up by the binder removal (debinding) and high-temperature sintering to afford the target ceramic article. This stage plays a crucial role in the production of pore-free equal-density ceramics at both industrial and laboratory levels.

The industrial usage of carbon nanotubes (hereinafter, CNTs) has experienced a tremendous expansion over the last decade. Due to advances in manufacturing technologies, CNTs become progressively cheaper and thus cost-effective for diverse large-scale applications [1]. CNTs are successfully applied to reinforce composites based on either metals or ceramics. Ceramic composites reinforced with CNTs characterized by diameters from sub-nm to several tens nm (and lengths from 1 μm to a few cm) are commonly referred to as ceramic matrix nanocomposites, or CMNCs. Properties and applications of CNT-reinforced CMCs have been a subject of numerous recent reviews [2–9]. Most typically, a CNT-type additive is introduced into a ceramic matrix to improve on its mechanical properties, in particular, hardness and fracture toughness [10,11]. The size and shape of the reinforcing component should meet specific requirements such as a length much larger than the

ineffective length and a sliding interface [12]. Alternatively, carbon nanotubes could be added to a ceramic matrix to enhance its flexibility, thermal or electric conductivity (see Section 7 for more details). In most cases, it is desirable to achieve distribution of the CNT component in the ceramic matrix as homogeneous as feasible. Due to intrinsically anisometric shape, the CNTs often tend to adopt a specific preferred orientation in the composite. This has to be either suppressed or deliberately employed by using an appropriate method for the preparation of the molding mass and subsequent molding.

The design cycle of novel types of ceramics or CMC starts with a preliminary exploration of ultimate achievable properties. Test batches of the material are elucidated in terms of practically useful properties to identify their most promising application fields. If the application fields imply mass production, the development of cheaper manufacturing technologies and multi-parameter cost optimization become important issues.

Materials with maximum density and minimum size of crystallites (constituting ceramic matrix in the case of CMCs) manifest best mechanical properties that are frequently required for modern applications. Such ceramics or ceramic matrices can be produced from highly disperse or nanosized powders using specially adjusted mild compaction and sintering methods avoiding uncontrolled grain growth.

Finely grained and highly dense CMCs reinforced with CNTs can be routinely prepared via high-pressure or high-heating-rate sintering that makes it possible to minimize the retention time of material at the highest temperature. These requirements can be fulfilled by implementing hot pressing (HP) or hot isostatic pressing (HIP), pulse electric current sintering (PECS), or pulse plasma sintering (PPS). These methods are widely used for the preparation of advanced CMCs reinforced with CNTs at the laboratory scale. Meanwhile, these methods require expensive complicated instrumentation. Large-scale production of reinforced CMCs demands more cost-effective technologies typical of technical ceramics manufacturing and powder metallurgy to keep pace with increasing CNT affordability.

Before the sintering stage, a molding mass should be converting into a green body. Three types of methods are most commonly used to achieve this goal. These include: dry pressing of ceramic powders, plastic molding, and slip casting. To improve on functional properties of the target CMC the compaction stage should afford a green body with maximum possible mean density and density homogeneity but emergence of a texture (ordered arrangement of particles) is to be avoided in most cases.

## 2. General Description of Compaction Molding Methods Applicable to Ceramic Matrix Composites Reinforced with Carbon Nanotubes

A molding mass during compaction acts as an open system that is involved in energy, mass, and information exchange with the surrounding. This can give rise to the emergence of accumulative and dissipative structures [13]. In that state, the system becomes highly susceptible to diverse signals including internal (e.g., specific structures formed at previous stages of the system's evolution) and external ones (temperature, pressure, etc.). We note that the differentiation between accumulative and dissipative structures is formal. Further, one structural element can act as either accumulative or dissipative structure under specific conditions [14,15].

The compaction is accompanied by mutual flows of mass transfer within the molding mass. Quantitative characteristics of the process are strongly dependent on the binder content and compaction method applied. The shape of the mold and processes occurring at the interface between the molding mass and the mold are also important. The more complex the mold shape the more inhomogeneous is the field of friction forces and velocities of components upon filling the mold or passing through the molding unit (e.g., pipe extrusion, body of revolution rolling, green body machining, etc.).

Tribological properties of the molding mass, i.e., non-zero friction forces between the components (the so-called internal friction) and between components of the molding mass and mold walls (external friction) are of focal importance. The initially homogeneous distribution of components within a molding mass is disturbed due to local differences in internal and external friction coefficients. A non-spherical shape of particles is another reason for the inhomogeneous distribution of components

therein. The inhomogeneity is also resulted from inequivalence in the sizes and masses of all powder particles. It is known that the inhomogeneous distribution of components within a molding mass can give rise to an emergence of textures or ordered domains, which are commonly recommended to be avoided. All these general factors are valid for the compaction of CMCs reinforced with CNTs. Indeed, ceramic particles and CNTs are drastically different in terms of their tribological properties, shape, size, and mass of constituting particles. And these factors have to be taken into account. They cannot be suppressed completely but their negative effects can be and should be minimized.

We also note that the inhomogeneous particle distribution exerts positive effects in some specific cases. As a bright example of such a phenomenon, vibration-assisted casting of ceramic crucibles designed for metal melting can be mentioned. If the starting powder contains two fractions strongly different in size, finer granules are concentrated at the surface regions of the green body due to vibration. As a result, a denser and smoother ceramic article is produced. This can be regarded as self-organization and the rational use of the self-organization effects is a prominent trend in the design of innovative types of smart ceramics. Nevertheless, the preservation of a homogeneous component distribution is desired in the vast majority of cases. This can be routinely achieved via granulation. Furthermore, the granules should have a size distribution as narrow as possible. At the initial compaction step, the finely granular structure of the powder guarantees the uniform filling of the mold. But later on, the granular structure should be destroyed to ensure homogeneous density distribution in the resultant green body. The granulation approach has been successfully applied to the dry pressing but it is similarly promising for other compaction methods as well. X-ray computed tomography is the technique of choice to control density homogeneity in compacted bodies [16,17].

The compaction procedure is typically applied to a molding mass composed of pre-processed powder components and a temporarily introduced technological binder. It is absolutely necessary to pre-process CNT powders to improve their quality as a composite reinforcing modifier. Raw CNTs just after synthesis often contain admixtures and are aggregated to a certain extent, which require their purification, disaggregation, and surface modification as described in numerous publications [7–15]. The binder content largely determines the subsequent method of compaction. The molding mass preserves essentially powder form for the binder content up to 10–40 vol.%, which makes dry or semidry pressing most appropriate. Upon further increase in the binder content to 35–50 vol.%, the molding mass becomes plastic as plasticine, which makes plastic molding methods more efficient. Finally, at a binder content of as high as 50–70 vol.%, the molding mass acquires properties of a viscous liquid (which is thus called a ceramic slip). In such a case, green bodies are formed through slip casting. The borderline binder concentrations given above are conditional and can be varied over a wide range depending on the surface properties of solid granules, their size and shape, as well as exact type of the binder. The fundamental classification of methods used for the compaction and molding in ceramics technology is thoroughly discussed in many books [18–34].

## 3. Dry or Semidry Pressing

Let us start the systematic exploration of methods used for the compaction of powders with dry or semidry pressing. This method is applicable to systems containing no or a low content of a binder. Still, this results in dense and mechanically strong green bodies. It has to be stressed that this method is most popular for the industrial-scale production of many types of metal, ceramic, and composite species. The dry pressing is also the most cost-effective method thus it has to be used whenever it is feasible.

All methods of dry pressing can be classified into static and dynamic (pulse) ones. The term static implies that the external pressure on a molding mass is increased gradually at a relatively low rate. To the contrary, in dynamic methods, the external pressure is applied as pulses or shock waves and thus the rate of pressure raise is very high. The static methods of dry pressing include uniaxial pressing in sealed steel molds, cold isostatic pressing, and quasi-isostatic pressing, which are described in more detail below.

### 3.1. Static Methods of Dry Pressing

3.1.1. Uniaxial Dry Pressing in Sealed Steel Molds

The uniaxial pressing in sealed steel molds is the most widespread modification of dry pressing. It is most efficient to produce green bodies of a simple shape and relatively small height, such as cylinders, disks, plates, or bars. The method is easily automated and is characterized by a high throughput. Binder removal, or debinding, constitutes no problem due to its either total absence or low content. Nevertheless, mechanical stresses developing in the green body is the major drawback of the dry pressing. The problem escalates as compaction pressure increases. The stresses can transform into defects and emergence of cracks upon taking the green body out of the mold. As it has been mentioned above the method is especially efficient for small species of a simple geometrical shape with height smaller than diameter, or length and width.

The use of ultrafine and nanosized powders with a large excess energy result in a highly non-equilibrium mode of the process, which enables self-organization towards mechanically strong but highly porous framework structures. The mechanical energy transferred to the material upon compaction gives rise to emergence of dissipative and accumulative structures, according to the Le Chatelier–Braun principle [13]. These include local densification domains that tend to assemble into a common spatial framework [35]. The framework efficiently resists external load by channeling the incoming energy flow to the mold walls. As non-equilibrium accumulative structures, we may regard air bubbles captured inside the material, binder squeezed out of contacted granules, elastic deformation of powder particles, and transverse expansion of the mold sides. After pressure relief, they are transformed into elastic deformations of the green body mainly along the axis of compression and further to the emergence of overpressure cracks (laminations). The surface of newly formed cracks storing excess energy can also be regarded as accumulative structures. By understanding the origin of the processes, it becomes possible to balance their negative effects. For instance, a preliminary evacuation of the molding mass removes absorbed air and this gives rise to a substantial increase in the pressure, at which the overpressure cracks first appear. However, this preprocessing procedure is rather expensive and employed rarely.

Granules of the powder being pressed experience internal and external friction. The former concerns the friction between the granules whereas the latter includes the friction between powder granules and mold walls. It is external friction that makes the predominant contribution into the emergence of overpressure cracks. Thus, it is advisable to try to decrease it either by adding a special lubricant or using a special material of the mold characterized by a smaller friction coefficient with respect to the powder granules. CNTs as a component of a molding mass play a dual role. Their intrinsic elasticity escalates negative effects related to elastic deformations but the external friction effects would be suppressed due to highly lubricative properties of the nanotubes.

Several hints can be used to decrease the probability of overpressure cracks appearance. More specifically, the molding mass can be subjected to compression in a few distinct steps. First, it is pressed at one third of maximum pressure, then the pressure is raised to two thirds of the target value, and only after that sensibilization is the full pressure applied. Another hint implies the usage of a specially shaped mold. The pressing is performed in a cylindrical part of the mold. After completion of the procedure, the compacted green load is gently pushed out to the slightly tapered expanding conical part of the mold to relax mechanical stresses therein. Demolding by clamping the green body between two punches also works well. However, the clamping power has to be carefully dosed to avoid the green body's destruction.

A pre-processing of ultrafine powders into spherical granules is an efficient means to accelerate the compaction, achieve a homogeneous density distribution within the green body, and avoid overpressure cracks. To improve on the quality of granules even further the type and properties of binder should be adjusted as well. External friction effects can be reduced by adding special lubricants into the binder or on the interface molding mass–mold walls. The external and internal friction forces are drastically

distinct in nature and thus they make different and sometimes contradictory demands on properties of the lubricants. A special grease is often applied to the working surface of a mold prior to powder loading. The use of molds with ceramic surfaces instead of metallic ones also decreases the external friction since ceramics-ceramics friction coefficients are typically by far lower than metal-ceramics ones. However, such molds are substantially more expensive. In the case of molding masses for the production of CMCs reinforced with CNTs, both external and internal friction forces are decreased due to the presence of carbon nanotubes that are characterized by intrinsically low friction coefficients.

When a granulated powder is subjected to compaction by dry pressing the pre-existing granules act as accumulative structures rather than local densification domains of random size and shape emerging due to the self-organization. At early stages of the compression, trapped air easily escapes the molding mass through large intergranular pores. The strength of the granules has to be purposefully tailored such that they disaggregate at the maximum pressure to fill the pores and thus provide high and uniform density of the resultant green body.

To improve on the density homogeneity of a green body the biaxial pressing can be applied. In that case, external friction forces act in opposite directions and are applied only to a half of the molding mass rather than its whole (for the cylindrical geometry). Both punches exert pressure on the molding mass. It is the floating die that can be regarded as the simplest implementation of the biaxial pressing unit including a matrix freely floating being hung on springs. Hard rubber shims can substitute for springs in the laboratory practice. When the upper punch moves to increase the pressure it instantaneously stops and resumes moving together with the matrix towards the bottom punch. And this is equivalent to the upward movement of the bottom punch relative to the matrix. Upon multiple repetition this effectively provides biaxial pressing.

The use of pre-granulated powders, special binders, and lubricants that efficiently reduce both external and internal friction, taking precautions against the appearance of overpressure cracks, are further optimization options for the uniaxial pressing. This method is one of most cost-effective for the mass production of ceramic bodies of simple shapes especially when requirements to mechanical properties are not too challenging.

### 3.1.2. Dry Pressing in Collector Molds

The invention of collector molds first introduced by Khasanov and coworkers [36] has become a serious advancement in the technology of uniaxial dry pressing. Figure 1 compares a standard mold (Panel a) and two collector molds (Panels b and c). To realize the biaxial pressing mode with a standard mold the matrix 2 is mounted on a spring (floating die). A hard rubber or metal O-ring can be used in laboratory instead of the spring. After the first pressing step, the metal O-ring is removed, the mold is turned over and the pressing step is performed for the second time.

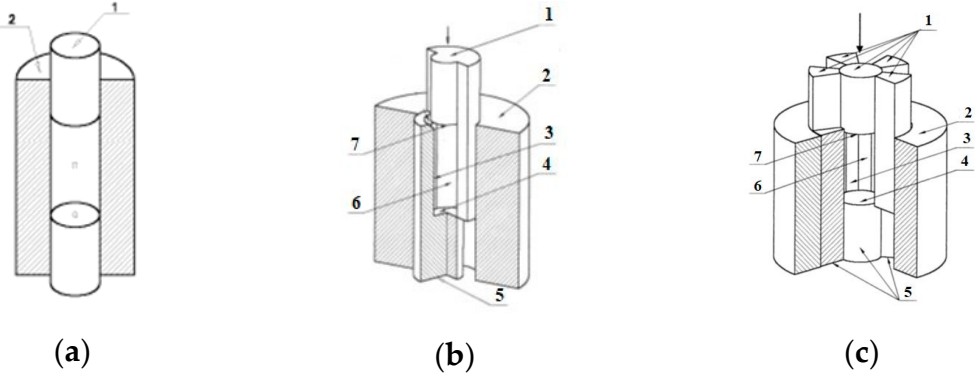

**(a)**　　　　　　　　**(b)**　　　　　　　　**(c)**

**Figure 1.** Cylindrical molds for dry pressing: (**a**)—standard mold (1—punch, 2—matrix); (**b**,**c**)—collector molds: 1, 5—independent punches; 2—matrix; 3 and 6—side punch surfaces; 4 and 7—pressing punch surfaces.

The collector mold design and the overall pressing procedure are more complicated. Figure 1, Panel b shows the simplest cylindrical collector mold. Two shape forming punches 1 and 5 are responsible for the pressure transfer. The element 2 that is essentially a matrix of the standard mold (similar to 2 in Figure 1, Panel a) suppresses off-axis drifts of 1 and 5. Both punches 1 and 5 (Figure 1, Panel b) are capable of moving independently with respect to each other and matrix 2. Elements 7 and 4 perform as active forming surfaces of the bottom and upper punches, respectively. Similarly, side surfaces of the punches correspond to elements 3 (side surface of 1) and 6 (side surface of 5). They constitute a passive shape forming surface separated along the pressure axis.

During the pressing step, external friction forces are applied to different parts of the passive shape forming surface (6 and 3) and characterized by opposite directions, although they are equal in absolute values. This tends to equalize the density throughout the molding mass.

In order to further reduce negative effects of external friction on density homogeneity the authors [37] suggested to divide the passive side surface into ten rather than only two independent parts, as shown in Panel c. The punches 7 and 4 perform the same shape forming function as elements 1 and 5. The matrix 2 again suppresses off-axis drifts. Elements 4 and 7 perform as active forming surfaces of the bottom and upper punches, respectively. Similarly, side surfaces of the punches correspond to elements 6 (side surface of 1) and 3 (side surface of 5). Such a division of the passive side surface of the cylindrical mold into multiple parts helps to compensate for mutual friction forces.

The number of slides moving in opposite directions can be increased even further to address requirements of specific technological tasks related to the high density homogeneity of a green body and low plastic shear deformation therein. Special designs of collector molds have been suggested for diverse shapes of green bodies and potential fields of applications, including conical gears, mills, impellers and turbine hydraulic pumps, components of toroidal, spherical, and helical shapes [38].

However, an increase in the number of independent slides has a limited potential due to complexity of their manufacturing technology. Instead of such a brute-force approach, the curved shape of the slides could be tested, which also increases the contact surface area. This can be achieved via twisting the slides around the pressing axis but symmetrically with respect to the height. It is also advantageous to divide slides by a helical surface with a coarse screw step rather than a simple plane [37–39].

The combination of helical collector molding with ultrasonic treatment exerts highly positive effects on the density uniformity of green bodies [40,41]. Using such an approach, green bodies demonstrating most isotropic shrinkage upon sintering have been prepared. The density gradients have been effectively reduced four-fold and two-fold along the height and in different cross-sections of the cylinder, respectively. For aspect ratios less than 0.7 (height/diameter), density gradients become equal in both directions. The use of ultrasound gave rise to green bodies that were better in terms of density gradients by factors of 6.5 and 2 with respect to ordinary dry pressing and collector molding, respectively. The ultrasound-assisted helical collector molding has proven very impressive efficacy in producing uniformly dense green bodies of very complicated shapes unfeasible by other dry pressing methods, in particular, spur gears.

The collector molding gets full advantage of using nanosized powders. The content of a solid phase can be up to 50 vol.% for a binder-free molding mass if nanosized powders are used. Advanced software algorithms have been elaborated to predict mold dimensions required to get green bodies of a strictly specified shape. It is important that uniformly dense green loads can be produced in the binder-free mode. This greatly facilitates the whole technology since the debinding step becomes unnecessary. However, the use of difficult-to-manufacture and thus expensive molds remains a major drawback of the method. Modern advanced techniques of metalworking and green body machining, application of gel casting, injection molding, and solid freeform fabrication (SFF) technologies can empower this method and make it more cost-effective and competitive. In our opinion, the method is highly promising for making green bodies of small sizes and complex shapes using CNT-reinforced CMCs.

### 3.1.3. Cold Isostatic Pressing

Cold isostatic pressing, or CIP, is a popular compaction method applicable to powder molding masses of complicated shapes [42]. The molding mass or preliminary compacted green body is placed into a bag made from appropriate elastomer (rubber, polyurethane, etc.) and transferred to an isostatic press [43]. The press applies a high pressure and efficiently compresses the elastic membrane and molding mass therein from all sides through the liquid phase serving as a pressure transfer medium. The density in the center is always lower than in outer parts even when an ideal sphere is being compressed. But a required density level is typically achieved at a lower external pressure as compared to the case of uniaxial pressing. CIP affords green bodies with a sufficiently uniform density provided that an appropriate molding mass is utilized [44,45]. The use of CIP virtually eliminates effects related to external friction, although internal friction remains important [46]. A correct design of elastomeric molds becomes also essential [47,48]. Notably, both the size and shape of an elastomeric mold typically differ from those of a targeted green body. Properties of powder and granules within the molding mass strongly affects characteristics of resultant green bodies, similar to the case of uniaxial pressing discussed above [49,50]. Pressures from ca. 20 MPa to 400 MPa and even up to 1 GPa in rare cases are typically utilized for CIP. After ramping the pressure to the maximum desired level, it is slowly relieved to the ambient value to allow relaxation of stresses. Then the compacted green body is unloaded from the mold. Ultrafine and nanosized powders are often evacuated prior to sealing. A starting powder molding mass or preliminary compacted green body can be placed into the bag for CIP. For the former case, it is usually difficult to control the exact shape and size of the resultant green body. That is why a preliminary compaction step is advisable. The partly compacted green body can be produced by any method, including the optional debinding step. The application of CIP in that case affords more advanced green bodies with improved mean density and density homogeneity.

There are two modifications of CIP, namely wet-bag and dry-bag. The former modification implies that the liquid phase is all around the molding mass. The requirement is not fulfilled for the latter modification. They are schematically shown in Figure 2. In the case of dry-bag CIP, the initial powder is loaded into the mold and the resultant green body is unloaded from it through the liquid-free space. Wet-beg CIP presses are often characterized by a large size: the high-pressure chamber can be as large as 1.8 m and the total height of the press can be up to 3.7 m. Many molds with green bodies of different sizes and shapes can be loaded into such a press to be compacted at once. Most typically, the elastomeric molds are disposable since they are destroyed upon unloading, although this rule is not absolutely strict.

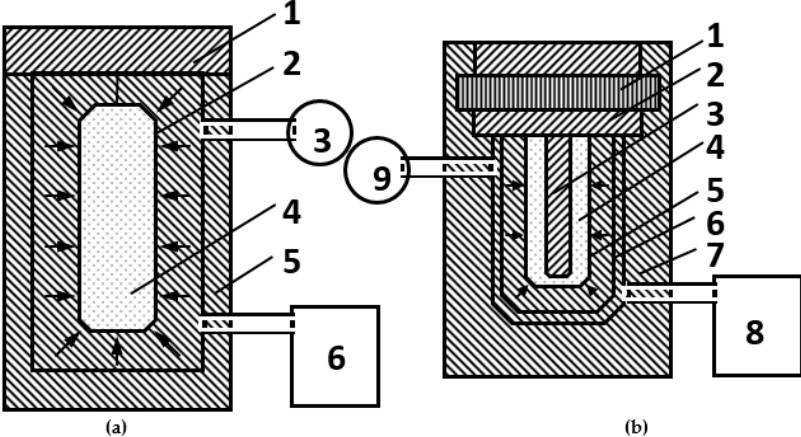

**Figure 2.** Schemes of CIP presses: (**a**) wet-bag 1—lid, 2—elastomeric mold, 3—pressure gauge, 4—powder, 5—high-pressure chamber, 6—compressor; (**b**) dry-bag 1—bayonet shutter; 2—cutter; 3—cavity-forming core, 4—powder, 5—elastomeric mold, 6—perforated limiter conforming to the green body shape, 7—container, 8—compressor, 9—high-pressure chamber.

The wet-bag CIP has a number of advantages, including the possibility to compact many green bodies of different sizes and shapes at once and a relatively high density homogeneity of the resultant green bodies. The most important drawbacks are relatively high costs and difficulties with the precise control over size and shape of the green body that typically requires finishing machining. A pressing operation takes a relatively long time from 5 to 60 min (which reduces overall performance) and is difficult to be automated (although some automation is possible).

In the dry-bag CIP, the elastic mold typically constitutes an integral part of the press and thus the green body can be taken out avoiding mold destruction. The method can readily be automated and implemented in the high-throughput mode. For instance, the mass production of automotive spark plugs is realized using this method. Meanwhile, the dry-bag CIP is inferior to wet-bag modification when a technology for manufacturing new ceramic parts is to be established. The wet-bag CIP is by far more popular at laboratory-scale production. The compaction stage can be followed by finishing machining in the case precisely shaped ceramic bodies are required.

CIP is well suited for the compaction of powder molding masses of complicated shapes using CNT-reinforced CMCs. High costs of isostatic presses and necessity to make and replace disposable elastomeric molds can be mentioned among key drawbacks of the approach. Still, the method is quite popular as an auxiliary compaction step to improve on the quality of green bodies prepared by other methods.

### 3.1.4. Quasi-Isostatic Pressing

The quasi-isostatic pressing makes it possible to realize the isostatic pressing mode using a standard uniaxial press without resorting to liquid, as in the case of CIP [47,51–63]. The isotropic pressure transfer is achieved due to a deformation of an elastomeric mold (made of rubber or polyurethane) upon uniaxial compression. The idea behind this method is schematically shown in Figure 3. When pressure is applied to the elastomeric mold it gets deformed and thus transfers the pressure in all directions. Indeed, this deformation is not genuinely isotropic as in the case of a pressure transfer medium. Therefore, this method is commonly referred to as quasi-isostatic pressing. There are density gradients on going from the punch surface to the peripheral parts of the green body, as in the case of CIP. The quasi-isostatic pressing is especially suited if the shape of target green bodies is not too complex. If necessary, the finishing machining can be applied after the compaction stage.

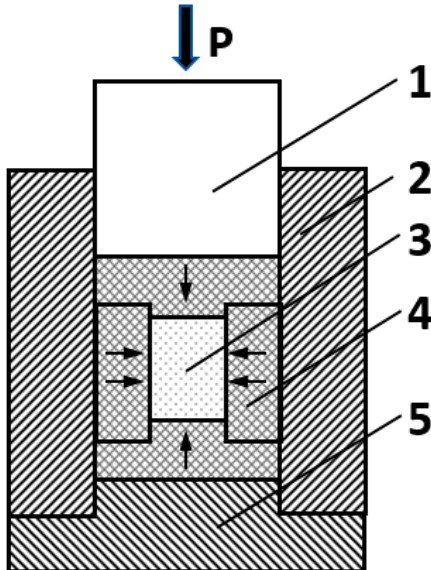

**Figure 3.** A schematic drawing of a cylindrical press for quasi-isostatic pressing: 1—punch; 2—matrix; 3—powder with a shape of a cylinder; 4—elastomer; 5—base.

For example, if it is necessary to mold a crucible, the bottom punch is made of an elastomer with the convex shape complementary to the crucible. It is centered on-axis at the flat metal punch. The powder molding mass is loaded from top and the pressure is applied also from top using a standard flat punch. The elastomer gets deformed under compression and thus properly molds walls of the crucible.

The quasi-isostatic pressing can be implemented with standard uniaxial presses and this is the main advantage of this method. The complexity of the design and manufacturing of elastomeric molds, difficulties with obtaining green bodies of exact sizes (especially, in the case of complex shapes) are unavoidable drawbacks of the method. Furthermore, it is not easy to be automated. And special precautions have to be undertaken in selecting proper binders and granulation pre-processing. Still, the quasi-isostatic pressing can find limited application for the compaction of green bodies with simpler shapes using CMCs reinforced with CNTs, as well as CIP.

### 3.2. Dynamic (Pulse) Methods of Dry Pressing

This group of methods include processes of compaction under the action of pulse external actions or shock waves generated by various sources [64–66]. Shock waves can be generated by an explosion of an explosive load or compressed combustible gas or liquid. Alternatively, an electrohydraulic discharge of a high-voltage capacitor through a water gap between two electrodes can be employed, or energy stored in a pulsed magnetic field. These energy bursts can be used for molding in the form of shockwave (or explosive) compaction, impact pressing, electroconsolidation, isothermal forging, as well as hydrodynamic and magnetic pulse pressing. Actually, the vibration and ultrasound-assisted pressing can also be classified as dynamic methods due to the periodic character of the external impact. Sometimes, these methods are placed into a separate group. But herewith, we will discuss them along with dynamic methods.

It is commonly precepted that it is the absence of external and internal friction that is the main advantage of the dynamic molding methods. The process just proceeds too fast for friction forces to develop. Furthermore, dynamic methods often afford green bodies with a higher mean density and density homogeneity with respect to their static counterparts. Nevertheless, they possess serious drawbacks as well. Their implementation requires highly specialized equipment and strict adherence to safety requirements. Instrumentation and first of all molds should be characterized by enhanced mechanical strength to survive shock wave impact of above 1 GPa at maximum. To fulfill such a requirement, this method is typically applied to molding masses of small sizes (below 100 mm). It is generally recommended to degas the molding powders as the pre-processing step (evacuation and heating). Ultrafine and nanosized powders should be carefully disaggregated. Green loads resulted from dynamic molding always exhibit unrelaxed stresses, which constitutes a serious technological problem. It is routinely solved by applying an additional temper annealing step prior to unloading from the mold. This degrades overall productivity and obstacles efficient automation.

#### 3.2.1. Magnetic Pulse Compaction of Dry Nanosized Powders

The magnetic pulse compaction that is one of simplest dynamic methods was initially developed for nanosized powders [67–70]. The powder is loaded into a mold and pressed under the action of mild pulsed compression waves with durations of 10–500 μs and amplitudes of up to 5 GPa (upon repetitive usage). Both external and internal friction forces are substantially reduced in that mode, which makes it possible to get green loads with a very high density even starting from a nanosized powder.

The energy primarily stored in a capacitor is converted into the kinetic energy of the press punch by an electromechanical converter. The mechanical impetus is generated due to the interaction of pulsed magnetic field with a conductive surface of a concentrator. The concentrator coupled with a punch is driven by the diamagnetic effect that tends to push a conductor out of the magnetic field. A typical design of the apparatus for magnetic pulse compaction (sometimes also called pulsed magnetic compaction) driven from top is schematically shown in Figure 4. To withstand periodic

mechanical loads, the concentrator has to possess sufficient strength and that is why it is typically bulky in size. The concentrator actuates the upper punch that compresses molding mass. Such a scheme corresponds to the uniaxial compaction. There could be two concentrators both from top and from bottom in order to double the compaction pressure and thus implement the biaxial compaction mode.

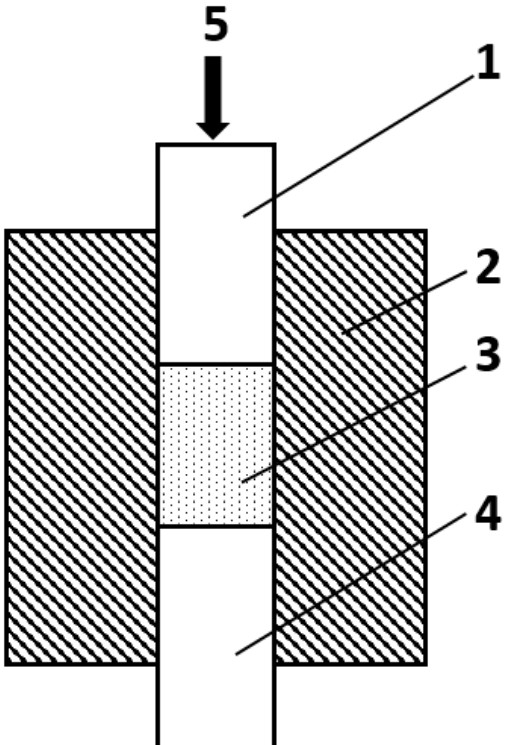

**Figure 4.** Schematics of magnetic pulse compaction with pressure applied from top: 1—upper punch, 2—matrix, 3—molding mass, 4—bottom punch, 5—mechanical load transferred from the concentrator coupled with the upper punch.

Record-high densities of green loads as high as 70–80% of theoretical values can be routinely achieved with the biaxial magnetic pulse compaction [71].

A heat is liberated in the regions of contact and mutual slippage of nanosized particles. Due to this, the hot mode of dynamic compaction can be realized by certain optimization of external shock waves. This mode is feasible for nanosized powders due to their high excess surface energy.

Nanopowders have to be carefully degassed by evacuation and annealing prior to the compaction stage since the amount of adsorbed gases can be as high as 20 mass.% in some cases. This pre-processing treatment is typically accomplished in moderate vacuum (~1–10 Pa) at 400–450 °C. The compaction is also recommended to be done in vacuo. After the compaction, the resultant green body should be subjected to the temper annealing step at 300–450 °C to relax mechanical stresses accumulated during compaction. The green body can be unloaded from the mold only afterwards. It is characteristic of magnetic pulse compaction that the material being molded accumulate substantial excess energy as structural defects. This stored energy decreases the temperature required for subsequent sintering by 200–300 °C, which is extremely advantageous since it suppresses the growth of crystallites in target ceramic species.

Nearly record-high densities of green loads can be achieved via this method. In particular, $Nd_2O_3$-doped yttria ceramics with a high optical transparency has been successfully prepared using magnetic pulse compaction [72]. Only explosive compaction can be regarded as a competitive alternative to magnetic pulse compaction but it is technically challenging. It would be sufficient to mention the need in use of explosives.

A special modification, namely radial magnetic pulse compaction, has been elaborated to mold rod- and pipe-shaped green bodies from nanosized powders. It relies on the impact of pulsed magnetic field exerted onto conductive cylindrical shells [68,73,74]. A sealed thin-walled copper shell with a shape of a tapered cylinder converts energy of the pulsed magnetic field into radial compression. The radial magnetic pulse compaction can be realized either in the electrodynamic Z-pinch or induction $\Theta$-pinch modes [75]. In the Z-pinch mode, electrical current runs through the shell filled with the molding mass and the shell is directly plugged to a pulsed current generator. In the $\Theta$-pinch mode, the shell has no electrical contact with the pulsed current generator but is inductively coupled with a special inductor unit. The Z-pinch mode is especially suitable for molding high-aspect-ratio (i.e., long and narrow) green bodies. The $\Theta$-pinch mode is applicable to relatively short blanks due to limited inductor dimensions.

Inertia effects can be efficiently exploited in the magnetic pulse compaction [76]. The magnetic pressure can accelerate the shell walls to a high velocity when the powder density is sufficiently low. Then the shell walls start to exert pressure and decelerate the magnetic and stored kinetic energies are summed up (and the latter can exceed the former several-fold).

A typical procedure for molding rod-shaped bodies includes several steps. First, the nanosized powder is loaded into a thin-walled copper pipe. Then the pipe is capped from both sides. For degassing, the pipe with powder is sealed with plastic gaskets placed between the caps and the pipe. The loaded and pre-processed pipe can be kept under ambient conditions for some time without severe precautions. For the compaction stage, the pipe is electrically connected to a capacitor coupled with a pulsed current generator.

Similarly, for molding pipe-shaped bodies, powder is loaded into a hollow space between the copper pipe and a central rod made from a hard material [77–79]. The thinner the walls of the pipe to be compacted, the more complicated is the task of dense and uniform filling of the cavity available.

The magnetic pulse compaction has been successfully applied to manufacturing thick-walled pipes from metal-ceramic composites to be used as hydraulic cutting nozzles. Thin-walled pipes as special design elements for fuel cells have also been manufactured. The use of current pulses with excessively high amplitude can destroy the green load, which is especially harmful for thin-walled pipes. According to theoretical simulations, the current of 0.8 MA is yet insufficient to get a green body of acceptable density, but destructive shock waves are already formed at 1.5 MA [80]. That is why a pipe-shaped green body (105 mm length, 18.7 mm outer diameter, and 0.9 mm wall thickness) from yttria-stabilized zirconia (9.8 mol.% of $Y_2O_3$, 15 nm mean particle size) was compacted to a relative density of 42–61% at a pulsed current of 1.0 MA. The subsequent sintering at 1360 °C (which is lower than typical temperature required for µm-sized powders by 200–300 °C) afforded a final ceramic pipe (83 mm length, 14.3 mm outer diameter, and 0.65 mm wall thickness) with a relative density of 97–99% and mean crystallite grain size of 200–300 nm. A solid oxide fuel cell (SOFC) unit assembled from such pipes demonstrated performance 1.5 times superior to industrial analogues [81].

The magnetic pulse compaction holds a special promise for the technology of CNT-reinforced CMCs. Any of basic shapes, i.e., disks, bars, and pipes, can be effectively compacted. Typically, binder-free molding masses are used, which eliminates unbinding-related problems. However, special efforts are to be undertaken to ensure homogeneous mold filling with the powder, which is especially challenging for ceramic-forming particles with carbon nanotubes anchored to the surface. A granulation pre-processing step prior to mold loading is generally recommended. For instance, the use of a ball mill to get round-shaped granules.

The presence of CNTs in the molding composition can increase stresses within the compacted green bodies, which can lead to appearance of cracks. This can be avoided by reducing the targeted relative density of the green body. The aforementioned work on thin-walled pipes demonstrated that the relative density of 45–55% at the green body stage is well sufficient to finish up with high-quality dense ceramics after sintering.

### 3.2.2. Vibration Compaction

The vibration pressing is often applied as an intermediate densification stage for ceramic and metal powders [82–85]. Vibration strongly reduces both internal and external friction within the molding mass upon compaction, which makes it possible to achieve relative densities of up to 90% of theoretical maximum. A molding mass can be either binder-free or contain a certain of amount of a binder. In the case of molding masses with a binder, one cycle of vibration compaction routinely gives rise to an increase in density by 6–12%. A scheme of the overall process is depicted in Figure 5. The vibration can be applied to either one or both punches or to the matrix.

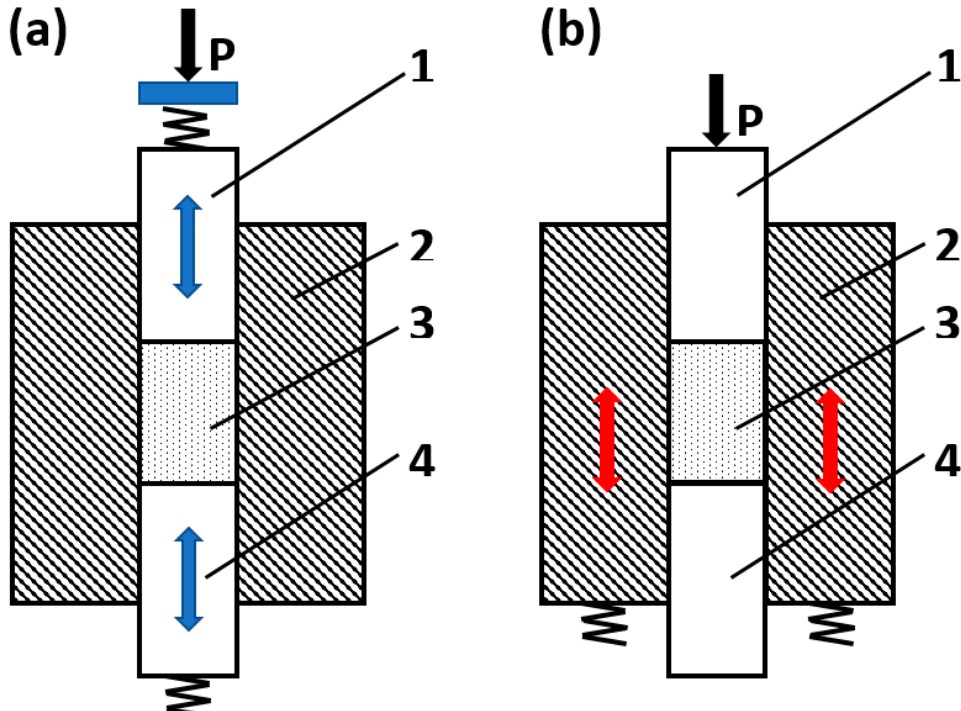

**Figure 5.** Schematics of vibration compaction. (**a**) The vibration is applied to both upper and bottom punches; (**b**) the vibration is applied to the matrix: 1—uper punch, 2—matrix, 3—molding mass, 4—bottom punch.

It is advisable to evacuate molding masses composed of highly disperse powder to improve on the density homogeneity, especially for molds of a complicated shape. There are two distinct types of vibration compaction, namely plain vibration and vibration-shock compaction. General protocols of green body preparation including the vibration compaction stage are also different. These include vibration compaction followed by static pressing, vibration-assisted static pressing without retention at the maximum pressure, and vibration-assisted static pressing with a retention at the maximum pressure.

An optimum packing of powder particle upon the vibrational impact is usually achieved within 2–10 s, therefore the compaction procedure should not take longer than 30 s. An increase in the vibration compaction duration is actually harmful: it doesn't improve on the density of the green body but gives rise to an escalation of internal stresses and segregation of fractions differing in size, which degrades density homogeneity.

The vibration compaction is especially efficient for powders of hard materials, which resist traditional static pressing methods, such as metal carbides, nitrides, or borides. Relatively strong green bodies with low internal stresses and relative densities of as high as 75–85% can be routinely achieved by vibration-assisted compaction at 0.3–0.6 MPa.

Optimum frequency and amplitude of the vibration depend on the particle size of the powder being compacted. The wavelength and particle size should of the same order and thus the vibration frequency should increase with a decrease in the mean size. In particular, this means that ultrasound should be applied for the compaction of nanosized powders.

For molding masses of a relatively small volume, the optimum vibration frequencies for particle sizes of <1 μm, 1–100 μm, and >100 μm are 300 Hz, 200–300 Hz, and 100–200 Hz, respectively. An increase in the volume of the molding mass from 10 cm$^3$ to 100 cm$^3$ typically requires an increase in the treatment duration by a factor of 1.5–2.

For powders with smaller particles, the pressure should be also increased. However, an increase in the pressure suppresses vibrations. Powder particles of a strongly anisometric shape are more resistant to compaction. Furthermore, the compaction of polyfractional powders proceed more readily than that of single-fraction ones.

The vibrational compaction is capable of affording green bodies of a complicated shape with a high density homogeneity, which is an evident advantage of the method. Furthermore, equipment necessary for its implementation is relatively simple and affordable. Nevertheless, vibration is harmful for personnel and exerts negative effects on parts and base of the press. The life cycle of parts of the mold and press significantly shorten upon repetitive vibration impact. Components of a molding mass can undergo segregation according to their sizes or friction coefficients. Finer fractions would tend to concentrate at working surfaces of the punches. These effects often occur in molding masses for CNT-reinforced CMCs.

It is typically recommended to optimize exact vibration parameters (frequency, amplitude, power) for each molding mass being compacted. This requirement can hardly be met using serial vibration presses. A versatile apparatus should be equipped with a vibrator tunable over a wide parameter space, which is very expensive. More dedicated instruments with vibration parameters tunable over a narrow parameter range are by far more affordable. Therefore, the application of vibration presses for CNT-reinforced CMCs is justified when a mass serial production of articles with complicated shapes and smooth surfaces is targeted.

### 3.2.3. Ultrasonic Compaction

The ultrasonic compaction is the method of choice for manufacturing high-density uniformly structured green bodies mainly from nanosized powders [86–90]. As in the methods described above, ultrasound impact decreases both internal and external pressure. Typically, no binder is required, which eliminates problems related to debinding. The relative density of resultant green bodies can reach 50%, which is relatively high for the binder-free compaction mode as applied to nanopowders. It is important to say that the green body density is weakly dependent on ultrasound characteristics and is determined primarily by the static pressure applied. The ultrasonic treatment makes nanosized powder particles or their aggregates vibrate and thus the intensity of ultrasound treatment should be adjusted so as the amplitude of forced vibrations is of the same order as particle (aggregate) size.

Ultrasound generators are characterized by a specific frequency range and power. Both parameters should be tuned depending on the size of the molding mass and effect required. The ultrasonic compaction decreases the elastic aftereffect in the compacted green body roughly proportional to the ultrasonic power (typically, 2–3 kW).

The amplitude of ultrasound-induced vibrations decreases with an increase in distance from the source. It means that it would be possible to identify a specific distance from the vibrating mold wall, at which the amplitude of forced vibrations corresponds to the particle size for virtually any nominal ultrasound intensity. Upon gradual increase in external pressure, density of the molding mass increases non-uniformly. The maximum density occurs at the surface contacted with the punch, whereas it decreases with depth. Thus, any region of the molding mass experiences ultrasonic impact in a nearly resonance mode at different stages of the process. Partly ordered local densification domains emerge in

the molding mass upon the ultrasonic treatment. These structures play a crucial role in the subsequent sintering stage improving on the mean density and density homogeneity of resultant ceramic species.

The ultrasonic power is typically fed radially, i.e., evenly in the plane normal to the external pressure. An ultrasonic compaction press is shown schematically in Figure 6. Special mold designs make it possible to transform radial vibrations into longitudinal ones parallel to the pressing axis. The external pressure effects are similarly suppressed irrespective of the ultrasound direction. But radially directed ultrasound improves density of the resultant green body since the vibrations penetrate into the bulk of the molding mass and affect internal friction as well. A combined application of ultrasonic compaction with collector molds is highly efficient (see above).

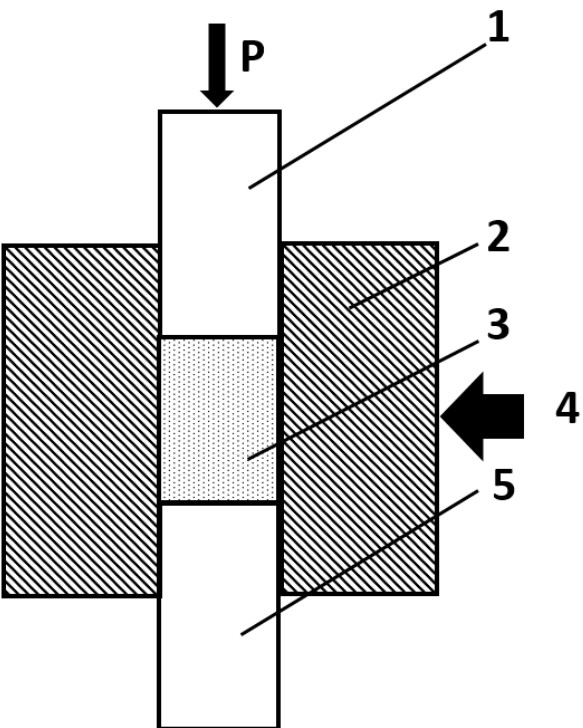

**Figure 6.** An ultrasonic compaction mold with a radial supply of ultrasound: 1—upper punch, 2—matrix, 3—molding mass, 4—direction of ultrasonic impact (generated by an ultrasonic spot head with the exponential concentrator, magnetostriction oscillator or another type of generator), 5—bottom punch.

High-quality optically transparent $Nd^{3+}$-doped yttria ceramics has been successfully produced using the ultrasonic compaction [91–93]. The ultrasonic power supplied was 3 kW for pressure values of 240 MPa, 480 MPa, and 720 MPa. The sintering of compacted green bodies was performed at maximum temperature of 2000 °C and heating rate of $V_T = 5$ °C/min. The ceramic bodies were kept at the maximum temperature for 30 h and then cooled down. The maximum optical transparency was observed for a green body compacted at the minimum pressure of 240 MPa. This means that the specific spatial structure of local densification domains is already formed at relatively low pressures applied. The emergence of such structures can be hampered or completely suppressed at higher pressures.

To the best of our knowledge, the ultrasonic compaction has never been employed for CMCs reinforced with CNTs so far, which could be a task for the future. Potentially, the ultrasonic compaction can afford highly dense and uniform green loads in the binder-free mode. But this method suffers from the necessity to use ultrasonic generators (that are to be adjusted to each specific sample) and special molds to channel the ultrasound into the sample. Moreover, the life cycle of molds is drastically reduced if the ultrasonic compaction is applied to abrasive powders.

## 4. Plastic Molding Methods

The plastic molding is similar to a some extent to dry pressing. Typically, a plastic molding mass contains 30–60 vol.% of a liquid binder in addition to a ceramic-forming powder. Important issues related to rheology of disperse systems as applied to ceramic technology have been discussed by many authors [94–103]. The list of practical binders includes aqueous solutions of polymers (in particular, ionogenic cellulose derivatives, alginates, and acrylates) or diphilic molecules (e.g., polyvinyl alcohol or diethylene glycol) [104,105], highly viscous organic liquids (either polymers or monomers or their solutions in organic solvents polymer) [106,107]. In the case of aqueous polymer solutions, the layer-by-layer adsorption of the polymer binder on the surface of powder particles is very important from the technological perspective [108,109].

The size and shape of plastic structural elements in a molding mass evolve in a complex way upon plastic deformation. The size of the structural elements can either increase or decrease when external pressure is applied. If the size increases upon pressing, which is referred to as dilatancy, the mechanical strength of the green body can be improved by appling a higher pressure. The opposite property called thixotropy implies that both mechanical strength and viscosity of the molding mass decreases upon pressing. From the technological viewpoint, it would be an ideal situation when a molding mass manifests thixotropy at the stage of loading into the mold or passing through the molding unit but becomes dilatant at a later compaction stage. Mechanical pressure on a molding mass does not typically exceed 1–30 MPa during the plastic molding. The molding mass keeps its shape due to non-zero yield point.

The shape of the structural elements is nearly as important as their size. A non-isometric shape of powder particles promotes the formation of texture (ordered domains). Anisotropic plastic strength is characteristic of textured molding masses. The presence of relatively large structural elements can be inherited at later technological stages as density inhomogeneity of a compacted green body and sintered ceramic article.

Antiplastifying additives to the molding mass are sometimes used in the plastic molding. A powder with coarser grains can act as such an additive since it introduces specific stable structural elements into the molding mass. Antiplastifying additives increase viscosity and reduce plasticity of the mass in a controllable way thus enabling rational tuning plastic properties. Meanwhile, discontinuities of mass flow occur around the large non-deformable grains. This effect is promoted by low plasticity and insufficient amount of surrounding plastic mass at a high deformation rate. Further, these discontinuities can either be safely cured or evolve into pores and micro/macrocracks with a chance of even a catastrophic failure. It should be kept in mind that the cracks can appear not immediately after the compaction. They could occur later on at debinding or even after sintering.

For the plastic molding, even air bubbles trapped in the molding mass can act as antiplastifying additives. Air adsorbed on the surface of powder particles or dissolved in water is pushed out to boundaries of the structural elements upon their growth effectively reducing friction forces and strength. Although this can also give rise to emergence of cracks, the effect can be used to advantage. For a higher gas content, special triple liquid-gas-powder fluid mixtures can be utilized similar to the technology of metal casting.

A weakening of boundaries between coarse structural elements when the liquid binder is squeezed out can give rise to the emergence of specific discontinuities in the form of glide planes. These regions are characterized by lowered pressure, which promotes air bubbling. Air bubbles prevent curing these defects at later stages and thus the transformation of the discontinuity into a crack becomes more probable. The external friction against walls of the mold or molding unit decreases the rate of mass flow, which gives rise to gradients of velocities along the cross-section of the green body. It is a general rule of thumb for the plastic molding, the internal pressure should be greater than the external one [110].

The plastic molding is routinely applied to bodies of a complicated shape characterized by thin walls and high aspect ratios, in particular, pipes with non-cylindrical holes. The vast majority of

polymer suspensions and solutions are thixotropic. Only highly concentrated systems with a low stability are dilatant.

Reversible elastic and delayed deformations of plastic masses play an important role at low rates. Plastic masses that are prone to delayed reversible deformations are most attractive for the plastic molding.

Mineral composition, shape and size of solid particles, binder type, and content mainly determine the overall functional properties of plastic molding masses. The binder content should be carefully optimized for each specific case. In the case of water-soluble polymers, their affinity towards formation of hydrate layers atop particles' surface should be taken into account [110].

The number of close contacts between particles in a molding mass per volume unit, internal friction coefficient, and strength increase with a decrease in the mean size of the particles. To compensate for negative effects related to yield point, viscosity, plasticity, and Young's modulus, the binder content should be appropriately increased. This increases shrinkages of a green body upon debinding and sintering. For water soluble binders, their optimum content could be reduced to a certain extent by introducing electrolytes.

### 4.1. Extrusion Molding

The extrusion molding is typically realized using vacuum screw (auger) presses or, more rarely, piston (ram) extruders. A screw press resembles a meat grinder, whereas a piston extruder is similar to a syringe. The extrusion molding is efficient in making green bodies of a rectangular of circular cross-section (rods or pipes) or as a preliminary step for further stamping molding. During the molding, plastic structural elements are drawn along the axis of the mass transfer. The binder concentrates at boundaries of these elements and also acquires an elongated shape. As a result, elongated pores are formed upon debinding [111].

A screw vacuum press is a continuous operation high-throughput apparatus, which however requires a relatively soft molding mass as feed. Specific defects occur in the green body due to velocity gradients during screw-driven movement of the mass through the press body and molding unit (die). This primarily gives rise to an inhomogeneous distribution of components across the molding mass. The axial movement involving highly anisometric particles always results in a texture. Elongated and flat particles are oriented parallel to the extrusion axis with their longest dimension and flat side, respectively. These effects are especially prominent in molding honeycomb bodies. Furthermore, these effects can be deliberately utilized to induce practically important anisotropy of functional properties in fiber-reinforced composites, in particular, those containing CNTs. For instance, extrusion molding has been applied to mold textured green bodies from $\alpha$-$Si_3N_4$ reinforced with $\beta$-$Si_3N_4$ fibers (15%) [112]. This method is similarly promising for CMCs reinforced with CNTs. To improve on the homogeneity of mass flow a variable screw step decreasing towards the press head could be applied. The use of double or multi-wing end auger is another potent design feature.

Piston extruders are used to manufacture critical parts, such as honeycomb structures. They provide a mass flow essentially uniform along the cross section. Using such presses, great forces can be applied to a molding mass avoiding its rotation. However, such an apparatus can operate in the periodic mode exclusively.

Options to optimize the operation of a piston extruder include adjustment of head and die size, the use of tapered dies, application of special lubricants to the head and die. The internal and external friction forces can be further reduced under ultrasonic assistance supplied via special oscillating heads or another type of inserts.

### 4.2. Stamping Molding

Stamping is essentially a shaping and sizing technological operation applied to plastic molding mass preliminary compacted with a vacuum auger press. A precompacted mass is loaded into a metallic mold and subjected to a pressure.

The plastic molding mass is virtually incompressible at practically applicable pressure values (below 1 MPa) and the operation gives no increase in density of the green body. Thus, the molding mass is taken in a some excess when loading into the mold. The surplus mass is removed during stamping through special drills in the mold.

Plastic molding methods are superior to dry and semidry pressing when applied to green bodies of a complicated shape. However, the appreciable amount of binder used poses certain technological problems related to debinding and affords green bodies with increased porosity. To improve on the density of the green body, an auxiliary CIP step is recommended after the full or partial debinding. In some cases, the uniform distribution of components is disturbed in green bodies manufactured with a screw press. Such a probability is to be taken into account when developing technology for CNT-reinforced CMCs since carbon nanotubes are characterized by a very low friction coefficient with respect to most base ceramic materials. These negative effects should be corrected for by purposefully optimizing the binder type and molding regimes. Alternatively, they could be emphasized to get tailored CMCs with unique anisotropic functional properties.

### 4.3. Injection Molding

The injection molding (sometimes also referred to as ceramic injection molding, or CIM) [113–115] represents a step forward in the development of extrusion molding as applied to plastic molding masses. This method was extended to ceramics technology from polymer industry, where it has been widely employed for thermoplastics molding for decades. Thus, the respective instrumentation is very similar. Screw presses nearly the same as for the extrusion molding but smaller in size are typically used for the injection molding. The binder content is usually 30–50 vol.% (most frequently, 40–50 vol.%) [115,116]. A scheme of vertically oriented injection press in shown in Figure 7.

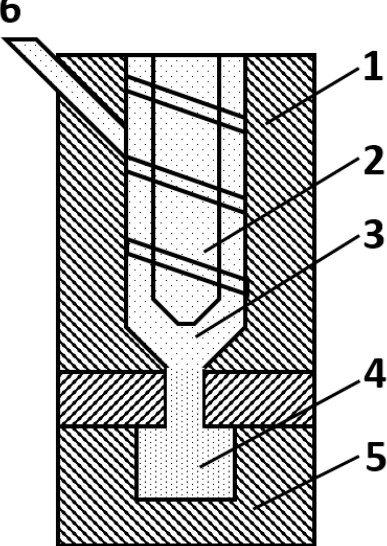

**Figure 7.** A scheme of a vertically oriented injection press: 1—heated housing, 2—screw, 3—molding mass, 4—green body, 5—mold, 6—molding mass loading.

The ever-increasing popularity of inorganic functional fillers gave birth to highly filled polymers that can be well regarded as green bodies for ceramic technology or powder metallurgy depending on the filler nature. A general term powder injection molding, or PIM, is widely used equally applicable to all three industries (polymers, ceramics, and powder metallurgy). Ceramic powders often manifest explicitly abrasive properties, this has to be taken into account when designing an injection press to avoid life cycle shortening due to rapid wear.

The injection molding is a cost-effective method for rapid (the molding cycle is only 1 min) serial production of small parts that are characterized by a complicated shape. The mass and linear size of green bodies rarely exceed 70 g and 0.5–5 mm, respectively. The method is suitable for producing articles with closed or through holes (>1.5 mm diameter) at strict tolerances <0.1 mm as well as individual parts with μm sizes [117,118]. Molding masses usually include powders with a mean particle size of 1–2 μm or nano-sized powders. The use of nano-sized powders enables manufacturing finely shaped articles with a characteristic spatial resolution of 10 μm and 2 μm for metal powders and ceramics, respectively [119]. Furthermore, a high reproducibility and surface quality are guaranteed [120]. Meanwhile, smaller particles increase viscosity of the molding mass (or feedstock, as it often called in the injection molding), which should be somehow compensated by an increase in the binder content. The latter should be used with a precaution since it gives rise to larger shrinkages at subsequent drying and sintering.

Compaction of really tiny articles, which is often referred to as microPIM, or μPIM, poses even stricter technological requirements [121]. A heated feedstock in the molten state is forced into a mold under pressure and then allowed to cool down prior to getting the resultant green body out of the mold. In thermoplastics technology, the molding polymer is typically loaded into a screw press in the form of granules. Ceramic molding masses could also be pre-granulated. As a binder, either a polymer or wax is taken and heated to 125–150 °C. Green loads afforded by the injection molding are usually sufficiently mechanically strong to admit subsequent machining. The high binder content gives rise to some challenges at the debinding step. The green body cannot be heated since it would result in binder melting and deformation. And thus, special low-temperature debinding protocols are applied.

The molten feedstock should contain as much solid phase as possible but at that possess sufficient fluidity, which are contradictory requirements and need to be balanced [122]. The search for novel water-soluble and environmentally friendly binders is currently underway [123].

Optimum binders for PIM and μPIM should meet the following set of requirements: maximum volume fraction of the solid phase, maximum mechanical strength of the green load, easy debinding preserving the defect-free structure of the green load and forming no toxic exhaust gases upon decomposition, low cost. The challenging debinding step is the most severe disadvantage in the application of PIM and μPIM methods to CNT-reinforced CMCs. A homogeneous component distribution achieved at the stage of feedstock preparation can get disturbed at the compaction or debinding stages. Meanwhile, the high strength makes it possible to mold green bodies of a simplified shape and then apply finishing machining.

## 5. Casting Methods

A molding mass suitable for casting should have a fluid slurry consistence owing to a high content of a binder (most typically, 50–70 vol.%, although lower contents can also be applied in specific cases). It is commonly termed slip [27–34]. Various liquids can be used as appropriate binders, in particular, water, aqueous solutions of polymers, organic solvents, or solutions of polymers in organic solvents. As with other methods, the content of a solid phase in the molding mass is to be maximized. However, highly concentrated suspensions become unstable against sedimentation of solid particles. That is why slips are usually kept under continuous stirring. The sedimentation stability of slips can also be improved by adding electrolytes, surfactants, or polymers. Slips should also be stable against aggregation. Aggregates tend to sediment much faster than individual particles.

In the case of slip casting, the friction between powder particles is minimum among all possible molding methods. It is highly advantageous for making uniformly dense green bodies but it fails sometimes. Uniform density of a green body is disturbed during compaction due to curved flow of the slip filling a mold of a complicated shape. This effect is especially severe for molding masses containing particles with different sizes and rheological properties. In particular, it is valid for CMCs reinforced with CNTs. This can be suppressed to a certain extent by maximizing the volume fraction of the solid phase in the slip without compromising its sedimentation stability.

To avoid deformation of the compacted green body upon unloading from the mold, the initial liquid suspension should be transformed into a mechanically strong solid. This can be achieved via several routes. The binder can be partially removed through porous walls of the mold. This route is followed in the case of aqueous slip casting into plaster molds. The removal of the binder increases the size of local densification domains in the molding mass and its viscosity but not to the full extent required for the safe green body unloading from the mold. The second route implies cooling. It is used in the case of casting from wax slips. A suspension in molten wax is poured into a mold (which is typically made from a metal), then it is rapidly solidified by cooling, and the resultant green body is taken out from the mold. The application of a similar approach to water-based solutions is risky taking into account the fact water expands upon freezing. Ice formation is accompanied by significant heat liberation, which cannot be efficiently withdrawn due to a low heat conductivity of ice. The third route relies on the polymerization of monomer-based suspensions. The gel casting also falls into this group. There are several methods to initiate polymerization. Thermally activated polymerization achieved by the mold heating is used most frequently. Chemical polymerization promoted by special catalysts is also used, although not as frequently. However, this approach is difficult to be implemented at the industrial-scale production. A reasonable compromise on the rate of the process should be found. The overall productivity degrades if the polymerization proceeds too slowly. But if the polymerization proceeds too fast, the molding mass could undergo solidification before uniform filling the mold or even already in the runner. Therefore, catalysts with a moderate activity are used and their activity is tuned by mild heating from the mold side. The second and third routes are widely applied to the plastic molding as well (vide supra).

The solidification of a molding mass can give rise to a redistribution of components therein. When a binder is partly removed through pores within the mold, traces of mass transfer channels remain in the green body. Anisometric particles are oriented so as to block the migration of the binder towards mold walls in the most efficient way. Plate-like particles are oriented with their basal plane parallel to the mold walls and whisker-like particles (including carbon nanotubes) are oriented with their main axis parallel to the walls.

The slip solidification (either due to freezing or polymerization) front moves from the mold walls towards its center. Solid particles would partly concentrate at this moving front, which would give rise to depletion of peripheral parts of the mold with solid particles. The front movement would induce the same preferred orientation of anisometric particles as in the aforementioned case of binder removal. An improperly filled molding mass can segregate into separate layers, which should be avoided. Such a situation could occur, in particular, upon casting from wax slips.

In the case of casting from aqueous slips into plaster molds the effective pressure of capillary water suction into gypsum can be as high as 0.1–0.2 MPa. Gypsum readily soaks water since its structure is characterized by a prominent hydrophilicity and high porosity due to a developed network of small open pores. Thus, the green body can be efficiently dried directly in the mold prior to unloading. This procedure does not damage the mold that can be reused multiple times. The process of binder removal can be accelerated by applying an external pressure to the mold (up to 4 MPa). This technology is referred to as pressure (or drain) casting. Water soaking and density growth is greatly accelerated at an elevated pressure. This strongly reduces the overall duration of the molding step, and the green body can be unloaded from the mold immediately without an auxiliary drying step due to its intrinsically low humidity. The molds can be made from porous polymers, metals, or ceramics in order to extend their life cycle in terms of reusability. Most popular polymer molds can survive as many as 50,000 repetitive casting cycles without degrading their performance. Furthermore, polymer molds are less expensive than metallic or ceramic ones. Plaster molds are reusable for not more than 80–100 times.

*5.1. Casting from Aqueous Slips*

The casting from aqueous slips is widely used for manufacturing thin-walled hollow and solid green bodies from either construction or technical ceramics [124–131]. For making solid green bodies,

additional doses of the slip are added to the mold several times. As a result, the green body precisely reproduces the internal shape of the mold. To the contrary, in the case of a hollow green body, the slip is poured into the mold and its excess amount is removed right after an appropriate wall thickness is formed. Typical casting procedures are schematically shown in Figure 8. The technology of casting from aqueous slips is relatively cheap. Costs for manufacturing porous plaster molds are low with respect to those made from metals or polymers [132]. In the technology of technical ceramics, this type of slip casting is used when a small production batch is required. It has to be taken into account that calcium ions tend to diffuse from gypsum through the slip into the green mold. Surprisingly, but this small amount can be sufficient to induce crystallization of amorphous $SiO_2$ during either sintering or subsequent routine exploitation of the respective ceramic article. The crystallization is accompanied by a large jump in specific volume, which gives rise to the emergence of cracks. Porous polymer molds are usually employed for the mass production instead of gypsum ones.

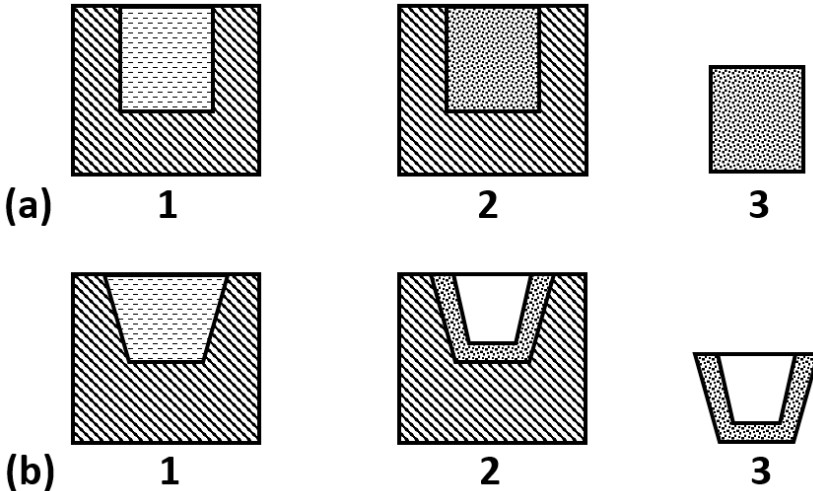

**Figure 8.** Schematics of casting from aqueous slips: (**a**) manufacture of a cylinder-shaped green body with several slip re-pouring cycles; (**b**) crucible manufacture with removal of an excess slip.

A careful optimization of the granular composition and prolonged grinding in water afford highly concentrated binding suspensions (HCBS) that preserve fluidity at an abnormally high content of the solid phase (up to 75–80 vol %) [133–136]. For a fluid system based on quartz glass, even higher content of 83% has been reached [137]. Owing to the presence of gel-like nanoparticles on the surface of ceramic-forming particles the resultant dried green body had a high strength. Pivinskii and co-workers used powders consisting of relatively large particles. The large particles provided fluidity of the molding mass and reduced shrinkages of the green body at the stages of drying and subsequent sintering. If finer particles are used the solid phase content would decrease, but the strength of the green body would increase. This would allow one to use a lower sintering temperature and finish up with a ceramic article of a higher density. The lower binder content would facilitate the debinding step. This method is highly promising for CNT-reinforced CMCs unless finely grained composite is required.

In the case whereby CMCs reinforced with CNTs are prepared from μm- or nanosized powders, their proneness to aggregation should be taken into account. Polymer molds or molds made from the same material as that being compacted should be used if even minor contamination by calcium is unacceptable.

### 5.2. Casting from Wax Slips

The casting from wax slips can be regarded as a modification of drain casting using a low pressure [137–146]. A paraffin- or wax-based slip heated to 70–100 °C is poured into a metal mold at a

pressure of 0.2–0.4 MPa and allowed to cool down to afford a compacted green body. The slip should not get solid before it fills uniformly the entire mold. Then the molding mass is evacuated to remove air. If the slip is supplied to the mold too fast it traps too much air, which gives rise to the formation of pores in the green body. A typical rate of slip supply into the mold is 1–2 m/s.

The casting from wax slips can also be used for compacting hollow green bodies, such as crucibles. The overall procedure in that case is similar to the casting from aqueous slips with the removal of excess slip. The slip is poured into a mold. It is important to provide an efficient heat sink from the mold. When walls of the crucible with a required thickness are formed, the excess slip is removed. Alternatively, the green body for a crucible can be made by immersing a metal part with the shape of the internal surface of the crucible into a wax-based slip kept at 60–90 °C. Due to an efficient cooling, a layer of the wax slip is solidified at the metal part. The resultant green body would have a desired crucible shape with relatively thin walls.

The technology of wax slip casting is similar to a some extent to metal casting and uses same tricks like chill molds. Continuous casting is used for manufacturing pipes and bars. The wax slip is fed into a cooled die kept at a certain temperature somewhat below the binder melting point. This allows the green body to retain its shape. Hot stamping can be used for making green bodies with relatively simple shapes. The slip is poured into the mold and kept where under a mechanical load until complete solidification.

A complex-shaped green mold can be prepared in several distinct steps. First, its simpler parts are molded an then they are welded together into the targeted article. The surfaces to be welded are heated to 60–70 °C, pressed together at a moderate force and allowed to cool down.

The casting methods are cost-effective in manufacturing small batches of green bodies irrespective of their size, also using CNT-reinforced CMCs. Both standard and nanosized powders can be used. Distortions of the slip structure upon filling the mold and solidification constitutes a technological problem (see above). The problem is especially severe if carbon nanotubes are used. Different densities and wettabilities of CNTs and ceramic-forming powders would result in their non-uniform distribution over the slip volume. These negative effects can often be suppressed by selecting proper stabilizers and increasing the solid phase volume fraction.

*5.3. Tape Casting*

The tape casting is an ad hoc method deliberately optimized for manufacturing thin ceramic films (with a thickness down to 10 μm. For that, a thin layer of a slip is poured onto a moving liquid-proof surface made, for instance, from Mylar or another similar polymer. The width of the substrate tape can exceed 1 m. The film cast in such a way is rolled up for the storage and further processing. Thin film ceramic articles find broad applications in modern electronics.

For the precise film thickness control, a doctor blade system is applied. A slip is poured from a reservoir onto a moving Mylar tape. The gap between the reservoir bottom (that is the doctor blade) and the substrate tape determines the thickness of the green body. The thickness is also affected by the movement speed, exact composition and content of the binder, drying conditions. The binder is of particular importance. It could be based either on water or a non-aqueous solvent [147–157]. Aqueous slips are cheap, non-flammable, and less toxic but they require more energy and time to get dry. Non-aqueous slips could be poisonous and flammable but they are more volatile. Furthermore, non-aqueous solvents are typically characterized by a lower surface tension, which reduces related defect formation. Chemical substances that are converted into ceramics upon annealing are sometimes added to slips for the tape casting [158]. The drying step is very demanding: a high uniformity both across the width and thickness should be ensured. For the tape casting, the same molding masses can be employed as in the gel casting (vide infra) [159,160].

If a molding mass contains strongly anisometric particles, in particular, carbon nanotubes, they would tend to orient themselves preferably with the main axis parallel to the film surface. Since the film thickness is usually small no depth gradient is anticipated although width gradients

are possible. Laminar CMCs reinforced with CNTs can be efficiently manufactured using the tape casting method.

*5.4. Gel Casting*

The gel casting [161–165] is a further development of casting method relying on the polymerization of monomer-based binder. A distinctive feature of the approach is that the water-soluble monomer undergoes polymerization via the gel formation. The use of monomer solutions with a low viscosity makes it possible to increase the content of solid powder in the slip. A green body can get completely solidified directly in a mold within 1 h with appropriately selected polymers. A highly homogeneous distribution of powder particles and gel gives rise to a small shrinkage of the green body upon debinding.

First attempts to adapt gel casting to ceramic technology dates back to late 1960-ies as surveyed in the seminal work by Ph. Colomban [166]. However, large-scale applications of the method started only in 1990-ies largely owing to activity of the Oak Ridge National Laboratory (ORNL) [167]. This method is suitable for the fabrication of complex-shaped green bodies, as with other casting methods. Standard casting equipment can be used for the gel casting, which makes it quite cost-effective.

Molds for the gel casting are typically made from either metals or polymers. Metal molds are characterized by a similar design as ones used for the injection molding. Aluminum-based alloys can be used instead of steel if the required molding pressure is not too high. Such molds are cheaper but still rather expensive especially when their shapes are too complicated.

Fugitive molds can be used for manufacturing green loads of especially complex shapes. They are made from special readily removable materials to facilitate green body unloading. This trick is similar to invest casting in metallurgy. Therein, molds are made from wax-based systems or urea. The latter cannot be applied to aqueous slips since urea is water soluble. For wax-based fugitive molds, the removal step implies either melting or dissolution. An important requirement to fugitive molds is that they should not deform upon mild heating required for the binder polymerization. Typically, the polymerization is accomplished at 100 °C, which is too high for standard wax systems that melt already at 50–70 °C. The melting point of a wax-based system can be increased by adding polyethylene, for example. Alternatively, the temperature of binder polymerization could be decreased. Fugitive molds can be reusable if the shape of a green body is sufficiently simple for easy unloading. But their stability against abrasive wear is usually low.

Batches of fugitive molds can be produced with a high productivity using either slip casting or injection molding. Molds afforded by this method are characterized by a high surface quality. Cavities of complex shapes can be stamped in fugitive molds using methods similar to investment (lost-wax) casting in metallurgy. The required cavity in a wax mold is shaped using an urea model that is dissolved in water afterwards. Fugitive wax molds are sufficiently mechanically strong to admit direct machining, including CNC machines, rapid prototyping, and 3D-printing [168–170]. These techniques are highly efficient in designing new molds but, unfortunately, they do not afford working mold surfaces with sufficiently high smoothness and cleanliness.

Special properties of binders are advantageous in the case of the gel casting since they retain fluidity at a very high solid phase content. And gel polymerization ensures a highly uniform distribution of solid particles and solvent throughout the green body. The solid particles are securely immobilized by the spatial gel network, which substantially facilitates debinding. First, a green load is desolvated. The remaining gel network dissipates stresses avoiding emergence of defects. Then the residual polymer is removed from the green body. This operation is straightforward due to the low polymer content and porous structure formed upon the desolvation. Highly accurate and reproducible shape, size, and mechanical properties of articles after sintering are achieved due to the impressive density homogeneity of green bodies molded by this method. Compacted green bodies are sufficiently strong to admit machining [171]. In particular, a green body made from 55 vol.% of $Al_2O_3$ and 3.4 vol.% of acrylamide/methylene bisacryalamide by the gel casting demonstrated a tensile strength of 30.8 MPa in a 3-point bend test [172].

A binder-forming monomer should bear at least two double bonds to enable cross-polymerization. Mixtures of two monomers are often employed. One monomer is responsible for the linear polymerization, whereas the second one acts as a cross-linking agent. A catalyst and an initiator should also be added to the molding mass immediately before the molding procedure. The initiator ignites the polymerization and the catalyst increases its rate and decreases temperature. Sometimes, both the functions are combined in one substance. A molding mass (slip) is usually evacuated prior to the gel casting. The vacuum applied should not be too high and the duration of the process should not be too long to avoid evaporation of binder functional components. Acrylamide was popular in recent years but its use was cancelled due to its intrinsic toxicity. Nowadays, methacrylamide (MAM) and methylene bisacrylamide (MBAM) are used in tandem most frequently [173]. The MAM:MBAM ratios lie typically in a range from 2:1 to 6:1 [161]. Polyethylene glycol dimethacrylate (PEGDMA) is also used in some cases instead of MBAM [161].

Polymers can be cross-linked via metal–ion complexation. In particular, this made it possible to develop robocasting microfabrication process requiring no human operator attendance. The technology is based solely on environmentally friendly polyvinyl alcohol [174,175]. E.-H. Kim, et al. [176] applied the gel casting to afford an extended series of samples including silica or zircon as ceramic-forming powders (60 vol.%); acrylic acid, N,N-dimethyl-3-oxo-butanamide, or ethylenecarboxamide as polymerizable monomers (3.3 wt.%), bisacrylamide as a cross-linking dimer (0.7 wt.%); homotaurine, ammonium sulfate, or ammonium peroxydisulfate as initiators (0.015 wt.%); and N,N,N'-trimethylethylenediamine or N,N,N',N'-tetramethylethylendiamine as catalysts. The total binder content was about 40 vol%. The authors successfully prepared ceramic cores with a high formability and green strength.

The development of new types of binders is currently underway. High-quality zirconia green bodies were prepared using phenolic resins that are very cheap but toxic [176]. The gel slip contained 42 vol.% of zirconia and 13 wt.% of phenolic resin. A high strength of green bodies can be achieved with water-soluble epoxy resins [177]. The solid phase content was increased to 48.9 vol.% without resorting to pH adjustment. Sialon ceramic obtained from a slip containing 44.5 vol.% of the solid phase had flexural strengths of 295.2 MPa and 281.4 MPa at room temperature and 700 °C, respectively. A ceramic helical spring with a rectangular cross-section has been prepared by the gel casting method using partly yttria-stabilized tetragonal zirconia (Y-TZP) [178]. Substantial efforts are being made in the search for environmentally friendly naturally occurring polymers for the gel casting, including proteins and polysaccharides [179–183]. In order to precisely reproduce the desired shape with alumina-based ceramics, the size of a targeted microarticle should be more than 30-fold larger than the size of powder particles [184]. Several new modifications of the gel casting have been suggested recently. In particular, gel casting has been effectively combined with the tape casting and injection molding. The latter got a name colloidal injection molding of ceramics (CIMC) [159,160,185–187].

Gel casting provides a high density and density homogeneities of green bodies. And this is reflected in mechanical properties of the resultant ceramic articles. A proper selection of environmentally friendly binders that admit high productivity and easy automation of the mass production remains a technological challenge. The gel casting requires no too complex equipment. Essentially, instrumentation very similar to that applied for the conventional casting may be employed. This makes the mass production of high-quality green bodies relatively inexpensive. This method is suitable for manufacturing green bodies of any size and shape, which are either dense or porous. The gel casting is very advantageous for molding green bodies using CMCs reinforced with CNTs.

## 6. Mold-Free Fabrication Methods

### 6.1. Solid Freeform Tape Casting Fabrication

The solid freeform fabrication (SFF) technology has experienced a tremendous progress in the recent decade [188]. There are a number of modifications within this general method, in particular: 3D-printing [189–192], stereolithography [193,194], robocasting [195], direct ink jet printing [196],

fused modelling deposition [197], etc. First, these technologies were successfully applied to make articles from polymers and metals. The adaptation of the SFF technology to ceramics appeared challenging due to brittleness, hardness, and high melting points of the most common ceramic materials. Nevertheless, the problems have been finally overcome owing to the bottom-up approach implying that a green mold is formed in a layer-by-layer manner. A digital CAD-generated model of the object to be fabricated is first sliced into thin cross-sectional layer representations. To get the full green body, layers are sequentially added up or deposited and fused to the previous ones. This technology makes it possible to obtain ceramic green bodies of ultimately complex shapes.

It has to be noted that presently these methods are not cost-effective for the mass production. However, they hold a quite competitive potential in manufacturing single articles of a unique shape. And this is a rather common demand. Let's imagine a situation. An engineer evaluates pros and contras of using a ceramic or ceramic composite part in a complex assembly. First, it is practical to make probe parts for their comprehensive testing. For an article of a complicated shape it is a common situation that the mold is many-fold more expensive than the targeted ceramic part. The fabrication technology optimization including the proper mold design taking into account debinding and sintering shrinkages is a multi-parameter resource-consuming task. Moreover, this challenge is bravely met by the SFF technology.

The need in single ceramic articles with a unique shape occurs repetitively in manufacturing orthopedic or dental implants in practical medicine [198]. In that case, the ceramic article has to be tightly customized to suit ideally to a specific patient and pathology type.

The poor surface finishing of ceramic green bodies is the major drawback of the contemporary SFF technologies. This strongly affects functional properties of the ceramic articles. An additional step of machining applied to either green load or sintered ceramic is laborious and costly. Therefore, improvements on the surface quality of ceramic green bodies made by the SFF technologies is one of most current technological problems. A breakthrough could be anticipated if the use of nanosized powders is adapted to the SFF methods. Huge efforts are put forth the rational design of novel binders that are environmentally friendly and afford stable molding masses with a high solid phase content. Stable and homogeneous molding masses are a prerequisite for making high-quality green bodies, especially in the case of CNT-reinforced CMCs. After all, an auxiliary green ceramic machining step can be added to the technological workflow if requirements to the surface smoothness and cleanliness are too strict.

*6.2. Green Ceramic Machining*

The green ceramic machining (GCM) as a technological operation can be formally classified as a modification of the mold-free fabrication. Moreover, it relies on the top-down approach, in contrast to the SFF ideology considered above, since a pre-fabricated article is finished to the required shape and size with machining.

Applications of GCM in ceramic technology has a long-standing history. It would be sufficient to mention green bodies of electric insulators that have been traditionally finished on lathes for years. However, the GCM technology has upgraded itself to the next level during the last decade [188,199,200]. More specifically, the computer-aided design and computer-aided manufacturing (CAD/CAM) tools have come into every day practice. Due to recent advancements of CNC machines, GCM has matured into a real alternative to molding. Primarily, the GMC is applied to fabricate one-of-a-kind ceramic articles, such as dental implants [201,202].

Machining of sintered ceramic articles is generally not recommended due to a high probability of emergence of surface microcracks degrading their mechanical strength [203]. On the other hand, the machining of green bodies requires by far less effort, and surface defects, when they appear, would be successfully cured at the subsequent sintering stage [204,205].

One may say that the GCM has already become an industry standard in ceramic processing [186]. The types of machining include drilling, turning, and milling [206].

Hardness and fragility of sintered ceramic articles cause severe problems for their machining. It makes this operation very expensive to the extent that finishing machining takes a dominant share of the net price of the article. As for a green body, the presence of a polymer binder imparts it a certain plasticity and reduces its brittleness. A green body should have a sufficient strength to admit machining. In most cases, the tensile strength of a green body is of the order of 5 MPa, which is not sufficient. The application of the CIP method together with resin binders increases the strength and facilitates machining. The maximum strength of green bodies as high as 10–30 MPa can routinely be achieved via gel casting [171,172,207–209].

A lot of dust is typically produced upon green body machining. A dust exhaust fan is usually installed next to the GCM cutting tool. It is advisable to collect the dust for further recycling.

Although diamond tools are preferable for GCM, many operations can be routinely accomplished with tools made from high-speed steel (HSS) [210]. HSS tools, however, suffer from rapid wear due to abrasive properties of ceramic powders. This poses negative effects onto surface quality of green bodies. But surface quality tends to improve upon subsequent sintering accompanied by mass transfer and defect curing. The GCM brings a green body to a size closer to the actually required one. This reduces the effective volume of finishing machining on the sintered article to stay within allowances and overall costs.

The non-contact GCM with lasers is a promising technology development direction [211]. The laser processing is applicable to green bodies with the solid phase content below 40 vol% [212]. At the solid phase content of 45 vol.% and higher sintering shrinkages become unacceptable. The material was gradually layer-by-layer evaporated with a $CO_2$-laser. The effective temperature of the process was 600 °C, which is much lower than that required to process sintered ceramics and thus cheaper.

The solid freeform fabrication (SFF) of ceramic green bodies and green ceramic machining (GCM) are very promising for the accelerated introduction of novel types of advanced ceramics into industrial practice. This first of all concerns various CMCs, including CMCs reinforced with CNTs. A simply shaped green body can be transformed into a complex shape by machining provided that it possesses sufficient density homogeneity. The solid freeform fabrication is especially suited to small production batches. If required, the surface quality can be further improved by an auxiliary machining step as applied to a dried or partly sintered green body.

The methods described above are powerful for molding ceramics from prospective inorganic materials that would come in broad use in the near future [213–218]. An innovative composite material has been reported very recently, which demonstrated an extremely high resistance against abrasive wear under the action of cutting tools and water jets [219]. The material consisted of a porous aluminum foam matrix and rows of alumina spheres contacting with each other through the matrix. The mean sphere diameter was 13 mm, whereas the aluminum foam was characterized by a mean pore size of 2.1 mm and interpore wall thicknesses between 30 μm and 1000 μm. The net density of the material was only 15% of standard steel. The astonishing wear resistance was due to a local resonance. The convex shape of spheres was expanded the hydroabrasive jet cross-section, reducing its velocity by two orders of magnitude.

## 7. Examples of Recent Studies Devoted to CNT-Reinforced CMC

In the last section, we would like to outline selected examples of CMCs reinforced with CNTs taken from papers published within last five years. The ceramic matrix composites were based on various ceramic-forming materials, several distinct CNT types as well as content were used therein. These were molded using different methods among those described above. Different functional properties of the composites were the subject of deliberate optimization, ranging from mechanical to electrophysical ones, according to their envisaged application fields. These examples are summarized below in Table 1.

**Table 1.** Examples of ceramic matrix composites reinforced with carbon nanotubes.

| # | Matrix | CNT Type | CNT Content | Molding Method | Property Being Optimized | Application | Ref. |
|---|---|---|---|---|---|---|---|
| 1 | $\alpha$-Al$_2$O$_3$ | MWCNT | 3 vol% | Dry pressing | Hardness and toughness | Biomedicine, aerospace and automobile industries | [220] |
| 2 | SiC | MWCNT | 0–9 mas% | Cold isostatic pressing | Microwave absorption, reflection loss | Weather radars, Doppler and telephone microwave relay systems, stealth technologies | [221] |
| 3 | SiCN | MWCNT buckypaper | 50 mas% | Gel casting | Electrical conductivity | High-temperature electrical devices | [222] |
| 4 | ZrB2-SiC | MWCNT | 10 vol% | Hot pressing | Fracture toughness | Thermal protection systems of hypersonic reentry aerospace vehicles | [223] |
| 5 | SiOC | Short carbon fibers | 15–30 vol% | Extrusion molding (direct ink writing) | Porosity, compressive strength | N/A | [224] |
| 6 | SiOC | MWCNT + graphene oxide | 1–10 mas% | Gel casting | Electrical conductivity | Hight-temperature fuel cells | [225] |
| 7 | SiCN | Carbon fibers + short CNT or long CNT | 45–55 vol% | Gel casting | Thermal conductivity, porosity | High-temperature structural applications | [226] |
| 8 | $\alpha$-Al$_2$O$_3$ | MWCNT | 5 mas% | Gel casting | Indentation fracture resistance | N/A | [227] |
| 9 | $\alpha$-Al$_2$O$_3$ | MWCNT | 0.35 mas% | Cold isostatic pressing | Electric conductivity, vacuum tightness | Vacuum electrophysical installations | [228] |
| 10 | $\alpha$-Al$_2$O$_3$ | MWCNT | 1.5 mas% | Hot pressing | Flexure strength, fracture toughness | Biomedicine, aerospace and automobile industries | [229] |
| 11 | SiCN | Aligned carbon nanotubes | 60 vol% | Gel casting | Flexibility | Thermal protection system and battery materials | [230] |
| 12 | $\alpha$-Al$_2$O$_3$-ZrO$_2$ zirconia toughened alumina (ZTA) | SWCNT, MWCNT | 0.1 mas% | Cold isostatic pressing | Fracture toughness, microhardness | Pump pistons, wear sleeves, spraying nozzles, steering nozzles, valve control discs, and as biomaterials for hip arthroplasty. | [231] |
| 13 | Si$_3$N$_4$ | Continuous multilayered carbon nanotube fibers | 7 mas% | Cold isostatic pressing | Oxidation resistance, conductivity, microwave absorption | Multifunctional structural composite materials | [232] |
| 14 | Mesoporous SiO$_2$ | MWCNT, including N- and B-doped | 2.5 mas% | Hot pressing | Hardness | Structural materials | [233] |
| 15 | $\alpha$-Al$_2$O$_3$ | SWCNT, MWCNT | 0.05 mas% | Cold isostatic pressing | Hardness, fracture toughness. | Structural materials | [234] |

## 8. Conclusions

Herewith, we surveyed general methods applied to molding ceramic matrix composites reinforced with carbon nanotubes. The CNTs are becoming less expensive and thus more affordable. They are considered to be promising for many new industries. Further, molding is the key technological stage in ceramic articles manufacturing, which determines their properties to a large extent.

First, it is advisable to perform a feasibility study on how the CNT-reinforced CMC parts would perform in their place within a complex assembly design. For that, a small batch of the parts should be afforded. A mold-free fabrication method could be applied as a cost-effective option. Alternatively, a green body of a simplified shape can be molded to be finished by the green ceramic machining. The green body should be characterized by high density homogeneity and tensile strength of above 10 MPa. These can be routinely achieved via gel casting, CIP, quasi-isostatic pressing or collector molding with subsequent machining and sintering. Field tests of the parts under realistic operation conditions would help to make a decision on prospects of their mass production.

Molding methods are rather diverse and numerous. No single method is optimum in all possible aspects. There are always specific advantages and drawbacks. With this paper, we have tried to give a brief introduction onto rational selection algorithm in the molding method—ceramics application coordinates. Large-scale manufacturing companies should be aware of this technological diversity as well as of recent trends and inventions in the field. Moreover, a potential customer would decide whether is more profitable either to rely on a renowned manufacturer offering the best price/quality products or to deploy production on their own.

**Author Contributions:** Supervision, V.P.M.; investigation, material systematization, writing, A.V.B. All authors have read and agreed to the published version of the manuscript.

**Funding:** This work was partly supported by the MUCTR internal grant.

**Acknowledgments:** The authors are grateful to the MUCTR grant for a partial support of the work.

**Conflicts of Interest:** The authors declare no conflict of interest.

## Abbreviations

| | |
|---|---|
| CAD | Computer-Aided Design |
| CAM | Computer-Aided Manufacturing |
| CIM | Ceramic Injection Molding |
| CIMC | Colloidal Injection Molding of Ceramics |
| CIP | Cold Isostatic Pressing |
| CMC | Ceramic Matrix Composite |
| CMNC | Ceramic Matrix NanoComposite |
| CNC | Computer Numerical Control |
| CNT | Carbon NanoTube |
| GCM | Green Ceramic Machining |
| HCBS | Highly Concentrated Binding Suspensions |
| HSS | High-Speed Steel |
| MAM | metacrylamide |
| MBAM | methylene bisacrylamide |
| MWCNT | Multi-Walled Carbon NanoTube |
| μPIM | Micro Powder Injection Molding |
| HIP | Hot Isostatic Pressing |
| HP | Hot Pressing |
| PECS | Pulse Electric Current Sintering |
| PEGDMA | Polyethylene glycol dimethacrylate |
| PIM | Powder Injection Molding |
| PPS | Pulse Plasma Sintering |
| SFF | Solid Freeform Fabrication |
| SWCNT | Single-Walled Carbon NanoTube |

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
