# Peer review of "Methods Used for the Compaction and Molding of Ceramic Matrix Composites Reinforced with Carbon Nanotubes"

_processes, doi:10.3390/pr8081004_

Round 1

Reviewer 1 Report

Authors present a very interesting review paper on the most important methods nowadays used for the compaction of molding masses composed of either technical ceramics or ceramic matrix composites reinforced with carbon nanotubes .

They made an extensive review on the subject, showing the different processes with their advantages and dissadvantages.

In my opinión the paper can be published in its actual form

Author Response

We thank the reviewer for the careful reading and the high appreciation of our work. No revisions have been requested.

Reviewer 2 Report

The manuscript gives a rather complete overview of pressing and shaping of the material, especially that containing a dispersion of carbon nanotubes, but the title is confusing and should be rephrased: not all CMCs are affected.

Authors should also consider the target properties and better explain the effect of CNTs on the properties. A CMC is generally defined as a ceramic matrix, in which a dispersion of another ceramic phase results in a non-brittle compound exhibiting a great work of fracture, and thus allowing the calculation of a reliable mechanical strength (A.G. Evans, J. Am. Ceram. 73 (1990) 187; K. K. Chawla, Composite Materials, Science and Engineering, 2nd Edition, Springer, 1998, New York). The size and shape of the reinforcement must meet specific requirements such as a length much larger than the ineffective length and a sliding interface (J. Wu, J. Raman Spectrosc. 28(7) (1997) 523). The addition of CNTs does not modify the mechanical properties but can change the electrical properties.

Another important parameter in CMCs is the homogeneity and isotropy or anisotropy of the dispersed phase which depends on the shape of the reinforcement and the shaping method. A paragraph on this point should be added.

The introduction should be reworded to clarify the objectives related to the introduction of CNTs into a ceramic matrix. The value of the different compaction and molding routes in relation to the target properties must be explained and the improvement of the properties demonstrated.

Information on the cost of the different techniques is welcome.

Note Gel casting routes are older than mentioned (see e.g. references in the review paper published in Ceram. Int. 15 (1989) 23).

Author Response

We are indebted to the reviewer for the careful reading of our manuscript and valuable comments. We have tried our best to incorporate the requested revisions into the revised version of the manuscript as describeb below. All changes made to the initial text are highlighted with yellow background (track changes mode has not been used).

Reviewer 2 comment: The manuscript gives a rather complete overview of pressing and shaping of the material, especially that containing a dispersion of carbon nanotubes, but the title is confusing and should be rephrased: not all CMCs are affected.

Our response: We have tried to improve further text in accordance with the reviewer's recommendations to put it into a tighter context. We would strongly prefer to keep the title unchanged and we believe it better corresponds to the revised text.

Reviewer 2 comment: Authors should also consider the target properties and better explain the effect of CNTs on the properties. A CMC is generally defined as a ceramic matrix, in which a dispersion of another ceramic phase results in a non-brittle compound exhibiting a great work of fracture, and thus allowing the calculation of a reliable mechanical strength (A.G. Evans, J. Am. Ceram. 73 (1990) 187; K. K. Chawla, Composite Materials, Science and Engineering, 2nd Edition, Springer, 1998, New York). The size and shape of the reinforcement must meet specific requirements such as a length much larger than the ineffective length and a sliding interface (J. Wu, J. Raman Spectrosc. 28(7) (1997) 523). The addition of CNTs does not modify the mechanical properties but can change the electrical properties. 

Another important parameter in CMCs is the homogeneity and isotropy or anisotropy of the dispersed phase which depends on the shape of the reinforcement and the shaping method. A paragraph on this point should be added.

Our response: we have completely reworked one paragraph in Introduction (lines 65-75): several recent reviews on CMCs reinforced with CNTs have been added (2-9). General goals of CNT introduction into ceramic matrices are formulated. The three references recommended by the reviewer have been also added. A concise discussion of issues related to homogeneity and preferred orientation of carbon nanotubes in CMCs has been added.

Reviewer 2 comment: The introduction should be reworded to clarify the objectives related to the introduction of CNTs into a ceramic matrix. The value of the different compaction and molding routes in relation to the target properties must be explained and the improvement of the properties demonstrated.

Our response: Instead of modifying Introduction, we have inserted a special section 7 at the end of the manuscript (line 1111-1120). The section now contains a Table with recently published examples of CNT-reinforced CMCs with most essential parameters, including "Property being optimized".

Reviewer 2 comment: Information on the cost of the different techniques is welcome.

Our response: This information would have indeed been very useful. But it is difficult to collect on a comprehensive and objective basis. Total costs depend upon numerous fine details. Some very qualitative estimates were in our initial text.

Reviewer 2 comment: Note Gel casting routes are older than mentioned (see e.g. references in the review paper published in Ceram. Int. 15 (1989) 23).

Our response: We thank the reviewer for noting this out. The respective part of the text has been reworded (lines 931-933) and the mentioned reference has been added.